



# Technical Note: Single-shell $\delta^{11}$B analysis of *Cibicidoides wuellerstorfi* using femtosecond laser ablation MC-ICPMS and secondary ion mass spectrometry

Markus Raitzsch[1,2,3], Claire Rollion-Bard[4], Ingo Horn[1], Grit Steinhoefel[2], Albert Benthien[2], Klaus-Uwe Richter[2], Matthieu Buisson[4], Pascale Louvat[4], Jelle Bijma[2]

[1]Institut für Mineralogie, Leibniz Universität Hannover, Callinstraße 3, 30167 Hannover, Germany
[2]Alfred-Wegener-Institut, Helmholtz-Zentrum für Polar- und Meeresforschung, Am Handelshafen 12, 27570 Bremerhaven, Germany
[3]MARUM - Zentrum für Marine Umweltwissenschaften, Universität Bremen, Leobener Straße 8, 28359 Bremen, Germany
[4]Université de Paris, Institut de physique du globe de Paris, CNRS, F-75005 Paris, France

*Correspondence to*: Markus Raitzsch (mraitzsch@marum.de)

**Abstract.** The boron isotopic composition ($\delta^{11}$B) of benthic foraminifera provides a valuable tool to reconstruct past deep-water pH. As the abundance of monospecific species might be limited in sediments, microanalytical techniques can help to overcome this problem, but such studies on benthic foraminiferal $\delta^{11}$B are sparse. In addition, microanalytics provide information on the distribution of $\delta^{11}$B at high spatial resolution to increase the knowledge of e.g. biomineralization processes. For this study, we investigated the intra- and inter-shell $\delta^{11}$B variability of the epibenthic species *Cibicidoides wuellerstorfi*, which is widely used in paleoceanography, by secondary ion mass spectrometry (SIMS) and femtosecond laser ablation multicollector inductively coupled plasma mass spectrometry (LA-MC-ICPMS). While the average $\delta^{11}$B values obtained from these different techniques agree remarkably well with bulk solution values to within ±0.1 ‰, a relatively large intra-shell variability was observed. Based on multiple measurements within single shells, the SIMS and LA data suggest median variations of 4.8 ‰ and 1.3 ‰ (2σ), respectively, where the larger spread for SIMS is attributed to the smaller volume of calcite being analyzed in each run. When analytical uncertainties and volume-dependent differences in $\delta^{11}$B variations are taken into account for these methods, the intra-shell variability is presumably in the order of ~3 ‰ and ~0.4 ‰ (2σ) on a ~20 μm and 100 μm scale, respectively. In comparison, the $\delta^{11}$B variability between shells exhibits a total range of ~3 ‰ for both techniques, suggesting that several shells need to be analyzed for accurate mean $\delta^{11}$B values. Based on a simple resampling method, we conclude that ~7 shells of *C. wuellerstorfi* must be analyzed using LA-MC-ICPMS to obtain an accurate average value within ±0.5 ‰ (2σ) to resolve pH variations of ~0.1. Based on our findings, we suggest to prefer the conventional bulk solution MC-ICPMS over the in-situ methods for e.g. paleo-pH studies. However, SIMS and LA provide powerful tools for high-resolution paleoreconstructions, or for investigating ontogenetic trends in $\delta^{11}$B, possibly due to "vital effects" during chamber formation.



## 1 Introduction

The boron isotopic composition ($\delta^{11}$B) of benthic foraminifera has been used to reconstruct deep-water pH (Hönisch et al., 2008; Rae et al., 2011; Raitzsch et al., subm.; Yu et al., 2010) and to estimate the Cenozoic evolution of seawater $\delta^{11}$B (Raitzsch and Hönisch, 2013). The underlying mechanism behind the boron isotope method lies in the constant equilibrium

fractionation of 27.2±0.6 ‰ between the pH-dependent speciation of trigonal boric acid and the tetrahedral borate in seawater (Klochko et al., 2006), where only the borate ion is incorporated into the foraminifera test (Branson et al., 2015; Hemming and Hanson, 1992).

However, while the number of studies on planktonic foraminiferal $\delta^{11}$B to estimate surface-ocean pH has rapidly increased within the last decade, deep-sea pH reconstructions based on benthic foraminifera are relatively rare. Possible reasons for

this might be the lower abundance of benthic foraminifera, compared to planktonic species, and a limited selection of species that truly record bottom-water, rather than pore-water conditions (Rae et al., 2011). Fortunately, there are two suitable candidates, *Cibicidoides wuellerstorfi* and *Cibicidoides mundulus*, that cover a relatively large oceanographic and stratigraphic range, and which have a high boron content of ~12-27 ppm (Raitzsch et al., 2011; Yu and Elderfield, 2007). Although their high [B] may partly compensate for the low abundance in the sediments, in many cases the availability of

enough specimens for $\delta^{11}$B analysis remains limiting.

Here, microanalytical techniques such as laser ablation multicollector inductively coupled plasma mass spectrometry (LA-MC-ICPMS) and secondary ion mass spectrometry (SIMS) can help to overcome the problem of sample limitation. These techniques have already been successfully used for a variety of biogenic carbonates to gain information on biomineralization processes or seasonal pH variations (e.g., Blamart et al, 2007; Fietzke et al, 2015; Howes et al., 2017; Kaczmarek et al.,

2015a; Mayk et al., 2020; Rollion-Bard and Erez, 2010; Sadekov et al., 2019). On the other hand, microanalytical analysis of $\delta^{11}$B is usually afflicted with larger uncertainties in terms of repeatability and reproducibility, as well as of natural $\delta^{11}$B heterogeneity within single shells and within a population. In addition, some recent studies using LA-MC-ICPMS suggest correction modes for measured $\delta^{11}$B values because detected interferences on the $^{10}$B peak, possibly due to scattered Ca ions from the carbonate sample, can result in large offsets from the expected value (Thil et al, 2016; Sadekov et al., 2019;

Standish et al., 2019), whereas in other studies this matrix-induced effect was not observed (Fietzke et al., 2010; Kaczmarek et al., 2015b; Mayk et al., 2020).

Also the reported analytical reproducibility for $\delta^{11}$B in biogenic carbonate using LA-MC-ICPMS differs considerably among different studies, ranging between ±0.22 and 1.60 ‰ (2σ =2 standard deviations), determined from repeated measurements of either a carbonate or glass standard (Fietzke et al., 2010; Kaczmarek et al., 2015b; Mayk et al., 2020; Sadekov et al., 2019;

Standish et al., 2019; Thil et al., 2016). As there is no standardized protocol nor a commercially available homogenized $\delta^{11}$B carbonate standard for determining the analytical uncertainty of LA-MC-ICPMS, this issue remains the most challenging task to compare the different labs and instruments. The most commonly used carbonate standards with well-constrained boron isotopic compositions are samples from a coral (JCp-1) and a giant clam (JCt-1), provided by the *Geological Survey*



*of Japan* (e.g., Inoue et al., 2004; Okai et al., 2004). However, for microanalytical analysis the standard is usually powdered
in a mortar and finally pressed to a pellet, which is produced individually in each laboratory, thus potentially resulting in
different heterogeneities (e.g., through different grain sizes or applied pressures) in each pellet. This issue is also true for
SIMS analyses, and the reported reproducibility is strongly linked to the in-house reference material used (e.g., Kaseman et
al., 2009; Rollion-Bard and Blamart, 2014).

In this study, we investigate how $\delta^{11}B$ in *C. wuellerstorfi* varies within single shells and between shells of a population of
tens of specimens. For this purpose, we used the femtosecond LA-MC-ICPMS and SIMS techniques and compared the
results with bulk-solution MC-ICPMS. Finally, we examine the size of population required for targeted $\delta^{11}B$ uncertainty
levels in paleoceanographic studies using LA-MC-ICPMS.

## 2 Material and Methods

### 2.1 Foraminifer samples

For this study, we used sediment samples from GeoB core 1032-3, taken in the Angola Basin on the Walvis Ridge at a water
depth of 2505 m. From a Holocene interval (6-8 cm, 5.6 ka), 23 pristine (glassy) shells of the benthic foraminifer species *C.
wuellerstorfi* from the size fraction >350 μm were picked and prepared for subsequent microanalytical analysis. Five large
specimens (>400 μm) were embedded in epoxy and polished down to a planar surface for SIMS analyses, while the
remaining 18 specimens were mounted on carbon tape for LA measurements. From these 18 individuals, two large tests were
analyzed for detailed chamber-to-chamber variability, while the remaining 16 tests were used to measure quasi-bulk $\delta^{11}B$ by
ablating large shell areas, preceded by measurements of the smaller umbilical knob area.

### 2.2 Secondary ion mass spectrometry

For the ion microprobe analyses, we used the same technique as described in Rollion-Bard et al. (2003) and Blamart et al.
(2007). Boron isotopic compositions were measured with the Cameca ims 1270 ion microprobe at CRPG-CNRS, Nancy,
France. A primary beam of $^{16}O^-$ ions generated using a Radio Frequency Plasma source (Malherbe et al, 2016) with an
intensity of 50 nA was focused to a spot of about 20 μm. A mass resolution of 3000 was used for B isotope analyses,
allowing the elimination of all isobaric interferences. Boron isotopes were analyzed in mono-collection mode using the
central electron multiplier. The deadtime of the electron multiplier was determined before the analytical session and set to 65
ns. A pre-sputtering of 120 s was applied before the analysis itself. The typical intensities of $^{11}B^+$ in foraminifer tests were
between 2000 and 4500 counts per second (cps), depending on the boron concentration. The analysis consists of 60 cycles of
10 s for $^{10}B^+$ and 6 s for $^{11}B^+$, respectively. The reference material was a calcium carbonate with a B concentration of 22 ppm
and a $\delta^{11}B$ of 16.76 ± 0.11 ‰, relative to the standard reference material (SRM) NIST 951 (WP22, value determined at IPGP





using the method of Louvat et al, 2014). The reproducibility, as estimated by multiple measurements of the reference material, was 2.48 ‰ (2σ, n=8), and is very close to the predicted 2σ uncertainty derived from counting statistics.

## 2.3 Femtosecond laser ablation MC-ICPMS

Boron isotope measurements were performed using a customized UV-femtosecond laser ablation system coupled to a Plasma II MC-ICPMS (Nu Instruments) at the AWI, Bremerhaven. The laser ablation system is based on a Ti-sapphire regenerative amplifier system (Solstice, Spectra-Physics, USA) operating at the fundamental wavelength of 775 nm with a pulse width of 100 fs and pulse energy of 3.5 mJ/pulse. Consecutive frequency conversion results in an output beam with a wavelength in the UV spectra (193 nm) and a pulse energy of 0.08 mJ. The short femtosecond pulses were shown to have major advantages over nanosecond pulses for a wide range of element and isotope ratios with respect to laser-induced and particle-size-related fractionation, thus enabling non-matrix-matched calibrations (e.g., Horn and von Blanckenburg, 2007; Steinhoefel et al., 2009). The sample and standard materials were mounted in an ablation chamber with an active volume of ca. 45 cm$^3$ ablated in a helix-mode scan at a speed of 2 mm s$^{-1}$ by using a laser spot size of ~40 μm. This technique allows producing ablation craters of almost any diameter, in this study ranging from ~80 μm for analysis of single-chamber to ~400 μm to cover whole shells. The aerosol was transported via a He gas flow (~0.5 L/min) and admixed with Ar gas (~0.5 L/min) before entering the MC-ICP-MS. The mass spectrometer was equipped with standard Ni sample and skimmer cones for dry plasma conditions. The radio frequency power was set to 1300 W. Boron isotopes were determined on Daly detectors, where high-mass D5 was used for $^{11}$B and D0 for $^{10}$B. Each measured sample $^{10}$B/$^{11}$B was normalized to $^{10}$B/$^{11}$B measurements of the glass standard NIST SRM 610 (δ$^{11}$B=0 ‰ NBS 951), using the Standard-Sample-Bracketing technique. The analyses were performed at low mass resolution (M/ΔM ~ 2000, 5‰), which was sufficient to resolve all interferences. We performed mass scans on the peaks of $^{10}$B and $^{11}$B for both gas blanks (laser off) and measurements on carbonate (laser on) (Fig. 1) to investigate possible effects by scattered ions of matrix elements as observed by some recent studies (Sadekov et al., 2019; Standish et al., 2019). For our set-up, we can exclude such matrix-induced effects, which is in line with Fietzke et al. (2010) and Mayk et al. (2020). Hence, there was no need to correct the raw LA data as done in the recent studies by Sadekov et al. (2019) and Standish et al. (2019). Before analysis, sample and standard materials were pre-ablated to remove potential surface contamination. Laser repetition rates ranged between 12 and 60 Hz to match the signal intensity between carbonate samples and standard material NIST SRM 610 (~300,000 cps). Torch position, ion optics and gas flows were optimized to gain maximum signal intensity and stability on $^{10}$B and $^{11}$B peaks. Each analysis consisted of 200 cycles with an integration time of 1 s and a prior on-peak gas blank measurement of 60 s, which was subtracted from the LA signal. After analysis, B was washed out for 120 to 180 s until reaching background levels before a new measurement was started. A typical blank had ~7,000 cps on $^{11}$B at the beginning of a session, but decreased to less than 3,000 cps during the course of a day. As signal intensity on $^{11}$B was aimed at ~300,000 cps, the signal-to-noise ratio was in the order of ~100. Any contaminated phase from partial ablation of clay infillings, indicated by dropping $^{11}$B/$^{10}$B ratios accompanied by rising [B], were excluded from further data treatment (Fig. 2).



Accuracy of boron isotope measurements was frequently checked by ablating an in-house carbonate standard and that was also used for SIMS analysis (i.e. WP22, Rollion-Bard et al, 2003). The average $\delta^{11}B$ of 16.49 ± 1.26 ‰ (2σ, n=20) for WP22 is very close to the bulk solution values ($\delta^{11}B$=16.60 ± 0.30 ‰ (2σ, n=6) measured at AWI, and $\delta^{11}B$=16.76 ± 0.11 ‰ measured at IPGP). As the measurement uncertainty is mainly dependent on the ablation time, we report measurement

uncertainties (2σ) for each $\delta^{11}B$ analysis as a function of analysis time, which was determined from multiple measurements of NIST glass standards and carbonate standards, and which is very close to the predicted uncertainty based on counting statistics (Fig. 3).

**2.4 Bulk solution MC-ICPMS**

After LA analyses, the 18 shells were carefully removed from the carbon tape and cleaned following the procedure outlined

by Raitzsch et al. (2018). Briefly, foraminifer shells were gently crushed under a binocular between two glass slides and transferred to Eppendorf vials. After the clay removal and oxidative cleaning steps, the samples were leached in 0.001 N $HNO_3$, and finally dissolved in 60 μL of 1 N $HNO_3$.

Prior to boron isotope analysis, we used the micro-distillation technique to separate B from the calcium carbonate matrix (Gaillardet et al., 2001; Misra et al., 2014; Raitzsch et al., 2018; Wang et al., 2010). The distillate was diluted with 400 μL of

140 0.3 N $HNO_3$. The B concentration of a small aliquot was determined using a quick (20 s) on-peak measurement of $^{11}B$ on Faraday cup H9 using a Nu Plasma II MC-ICPMS (AWI, Bremerhaven). The remainder of the sample was then diluted to yield a solution with a [B] of 3 ppb and concentration-matched with the SRM NBS 951 to within ±3 %.

For isotope ratio measurements, boron was collected on Daly detectors, where high-mass D5 was used for $^{11}B$ and D0 for $^{10}B$. Boron isotope data were measured in triplicate using the standard-sample-bracketing technique and reported in delta

notation normalized to SRM NBS 951:

$$\delta^{11}B_{sample} = \left( \frac{^{11}B/^{10}B_{sample}}{^{11}B/^{10}B_{NIST951}} - 1 \right) * 1000 \quad (1)$$

When 2σ of the mean derived from the triplicate was smaller than the long-term reproducibility (0.30 ‰), we report the latter as the measurement uncertainty. In addition, a small fragment of an in-house carbonate reference material WP22, used for our SIMS and LA-MC-ICPMS study, was cleaned and measured exactly the same way as the foraminifera sample to

150 obtain a bulk $\delta^{11}B$ value for comparison (16.60 ± 0.30 ‰). This value is almost identical to that measured at IPGP using the bulk solution ICP-MS (16.76 ± 0.11 ‰).





## 3 Results and discussion

### 3.1 Intra-shell $\delta^{11}$B variability

The results from SIMS measurements conducted on 5 large specimens reveal a high $\delta^{11}$B variability ranging between 4.6 and
155 6.8 ‰ ($2\sigma$) within single shells, based on 8 to 19 single spot analyses on each shell. A similar variability of 4.4 ‰ ($2\sigma$) on average is observed for measurements within single chambers (Fig. 4). The largest variation is observed in the central part (i.e. the juvenile chambers) of the shell, which may be attributed to the difficulty in distinguishing between the very small chambers in this part. Hence, we allocated these measurements to the umbilical "knob", which is also equivalent to the thick central part of the spiral side used for LA measurements. If measurements are averaged for each chamber (1 to 3 analyses per
160 chamber), the mean variability between chambers is 4.2 ‰ ($2\sigma$) (Fig. 4). The two specimens measured with LA also show variable $\delta^{11}$B, but with a much lower variation of ~1.1 ‰ ($2\sigma$), compared to the SIMS data (Fig. 4).

Here the question arises whether the difference in $\delta^{11}$B variability between the two methods is due to differences in analytical uncertainty or different scales of natural heterogeneity. If we consider an average uncertainty of ±0.7 ‰ for LA predicted by Poisson counting statistics (Fig. 3), intrashell variability is reduced from 1.1 ‰ to 0.4 ‰. As the $2\sigma$ measurement uncertainty
for SIMS is roughly ±2.5 ‰, the remaining difference in variability between SIMS and LA methods of ~3 ‰ is likely due to the different sampling volumes, and hence related to heterogeneous boron isotopic distribution in the test. While the spot size for the SIMS method is ~20 µm and ~1 µm in depth, the laser-ablated volume ranges from 80 to 100 µm in diameter (Fig. 4) and approximately 10 µm in depth. Consequently, the ~200 times larger volume analyzed by LA would reduce the $\delta^{11}$B variability detected by SIMS to ~0.3 ‰. Hence we argue that the "true" $\delta^{11}$B heterogeneity is scale-dependent and assumedly
in the order of ~3 and ~0.4 ‰ ($2\sigma$) on a ~20 and 100 µm grid, respectively.

To examine potential systematic trends in $\delta^{11}$B among successive chambers, we calculated the residual boron isotopic composition $\Delta\delta^{11}$B for each site within each shell by comparing the B isotopic composition of a single chamber $\delta^{11}B_{single}$ with the mean value of the shell $\delta^{11}B_{mean}$:

$$\Delta\,\delta^{11}B = \delta^{11}B_{single} - \delta^{11}B_{mean} \qquad (2)$$

The SIMS data suggest that $\Delta\delta^{11}$B tends to decrease from the penultimate chamber (f-1) towards chamber f-5 by roughly 4 ‰ (Fig. 4), whereas no systematic change exists between chambers f-6 and the juvenile chambers. However, it is compelling that also the LA results suggest a decreasing trend in $\Delta\delta^{11}$B from the final chambers towards chamber f-5 by more than 0.5 ‰, while in the earlier chambers no systematic change can be observed (Fig. 4). For both methods, Wilcoxon-Mann-Whitney tests suggest that the $\Delta\delta^{11}$B change between the final chambers and f-5 is statistically significant, but it should be
kept in mind that we are at the limits in determining ontogenetic trends, due to the relatively high analytical uncertainties of the LA and SIMS techniques. However, decreasing $\delta^{11}$B from the final chamber towards earlier chambers would be in line with the LA study by Sadekov et al. (2019) showing a ~2 ‰ decrease along the last whorl of *C. wuellerstorfi*. A similar pattern was also observed for B/Ca, with the highest value in the final chamber (Raitzsch et al., 2011; Sadekov et al., 2019), large-scalesuggesting a strong biological influence or kinetic (i.e. growth rate) effect on boron incorporation. An in-depth





discussion of biological/calcification processes is beyond the scope of this study, but the discovery of such high variability has implications for the use of $\delta^{11}B$-microanalytical techniques in paleoceanographic studies (e.g., Rollion-Bard and Erez, 2010).

Another notable feature derived from LA and SIMS is the somewhat elevated $\delta^{11}B$ of the umbilical knob, compared to the whole-shell $\delta^{11}B$. This is confirmed by supplementary ablation of the knob of individuals, which were used for whole-shell

analysis in section 3.2. On average, umbilical knob $\delta^{11}B$ was ~0.4 ‰ higher than the value derived from the larger ablated area (see inlet picture in Fig. 7), although this behavior is not systematic and was observed in only two thirds of the cases.

### 3.2 Inter-shell $\delta^{11}B$ variability

Apart from the seven specimens used for inspecting the chamber-to-chamber variability, 16 individuals of *C. wuellerstorfi* were laser-ablated using a large area of at least 300 μm in diameter to cover a major part of the spiral side, and subsequently

analyzed for the composition of the thicker umbilical knob using a smaller crater (inlet picture in Fig. 7). This way, we approached quasi-bulk $\delta^{11}B$ values for single shells. Together with the $\delta^{11}B$ medians from the two specimens described in the previous section, a total of 18 shells was used for determining the inter-shell $\delta^{11}B$ variability using LA-MC-ICPMS (Fig. 5). For SIMS analyses, the medians of single-spot analyses were calculated for each of the 5 shells.

The SIMS data reveal a huge spread of single-spot $\delta^{11}B$ across the 5 specimens (section 3.1), but the $\delta^{11}B$ values averaged for

each shell exhibit a narrower range between tests, with a median $\delta^{11}B$ of 16.08 ± 2.70 ‰ (2σ) (Fig. 5). In contrast, the single-site LA data across all 18 individuals show a smaller variation in $\delta^{11}B$ than the SIMS data, where the values averaged for each shell yield a median of 15.90 ± 1.62 ‰ (2σ). Both the average $\delta^{11}B$ measurement uncertainty for LA of ±0.9 ‰ (2σ) and the variation difference between foraminiferal shells and WP22 of ~0.4 ‰ suggest a residual inter-shell variability in the order of 0.4 to 0.7 ‰. Similarly, if an uncertainty of ±2.50 ‰ (2σ) for SIMS measurements is taken into account, the

remaining inter-shell variability is only ~0.2 ‰. Therefore, we estimate the "true" variability between shells of a population to be ~0.4 ‰, which is the same as the variation estimated for the intra-shell variability (section 3.1).

It is interesting to note that if only the large LA craters are considered, the mean $\delta^{11}B$ is 15.87 ± 1.78 ‰ (2σ), while it is 16.27 ± 2.75 ‰ (2σ), if solely the small LA craters are taken into account (cp. Fig. 7, right SEM image). As the volume of the large LA craters is ~3 times larger than the smaller ones, the resulting variability among means of 3 resampled small-

210 crater values is 1.59 ‰ (2σ), which is quite close to the 1.78 ‰ derived from large craters, and confirms our conclusion that the $\delta^{11}B$ variability is dependent on the scale at which it is measured.

### 3.3 Bulk solution $\delta^{11}B$

Both the SIMS and LA results reveal median values that match the bulk $\delta^{11}B$ of 15.99 ± 0.30 ‰ (2σ) measured in solution to within analytical uncertainties (Fig. 5). It should be noted again that the same specimens measured in solution had been

measured before by LA, ensuring that we compare different techniques based on the same set of samples. Similarly, the average $\delta^{11}B$ of 16.48 ± 1.26 ‰ (2σ) in the reference material WP22 determined with LA-MC-ICPMS is not distinguishable





from the bulk solution value of 16.60 ± 0.30 ‰ (2σ), which confirms the robustness of the LA technique, and also the SIMS results, as the median foraminifera values are identical for LA and SIMS techniques.

The δ[11]B values obtained from all three methods fit in with the calibration dataset for *C. wuellerstorfi* from the study by Rae
et al. (2011) (Fig. 6), and confirm that the boron isotopic composition in this species closely matches the one of borate of ambient seawater. Further, it proves that LA-MC-ICPMS and SIMS yield accurate results for δ[11]B, if the dataset is large enough to overcome the issues of intra- and inter-shell variability (~0.4‰), and analytical uncertainty of micro-analytical techniques (~±0.9 and ±2.5 ‰ for LA and SIMS, respectively).

### 3.4 Implications for paleoreconstruction studies

The large intra- and inter-shell variations in δ[11]B described in sections 3.1 and 3.2 raises the question whether microanalytical techniques such as SIMS or LA-MC-ICPMS can be used for analyzing δ[11]B in *C. wuellerstorfi* to reconstruct past deep-water pH. The SIMS method requires careful embedding of foraminifer shells in epoxy and polishing down to a planar surface, which precludes further processing for e.g. bulk solution analyses. However, because the size of the beam spot is small (20 μm or less), it is still possible to measure some other elemental and isotopic ratios at the same location on
the sample; e.g. the same foraminifera specimens were used to measure δ[18]O (Rollion-Bard et al, 2008), δ[11]B (Rollion-Bard and Erez, 2010), and δ[7]Li (Vigier et al, 2015). SIMS technique is very useful for biomineralization studies (e.g. Rollion-Bard and Erez, 2010), but for paleoreconstruction of deep-sea pH, where high precision is necessary, it may not be the most appropriate technique for routine downcore δ[11]B analysis. However, here we will inspect LA-MC-ICPMS as a potential tool for paleo-pH studies.

To attain information on the number of shells required for accurate LA analysis of δ[11]B to within a target standard uncertainty, we applied the 'combn()' function of the R package 'utils v3.4.4' (R Core Team, 2018). In this simulation, it is assumed that the entire population of *C. wuellerstorfi* consists of the 18 shells, for which we have measured δ[11]B both using LA and bulk-solution MC-ICPMS (Figs. 5 and 6). If from this sample only n individuals were available, we can calculate the standard uncertainty by generating all possible combinations of n shells taken from the entire population. The histograms of
resulting δ[11]B values averaged from n shells are shown in Fig. 7. For instance, if we would randomly pick four shells from this sediment sample, the analyzed δ[11]B would be accurate to within ±0.72 ‰ with a probability of 95 %. If we targeted a standard uncertainty of ±0.50 ‰, which is equivalent to a pH uncertainty of roughly ±0.1, we would need to measure 7 specimens with LA (Fig. 7).

Given that the analysis uncertainty of the same amount measured in solution is about ±0.3 ‰, bulk solution analysis appears
to be the more convenient technique for reconstructing paleo-pH. On the other hand, the LA technique may be useful for generating high-resolution records, where sharp pH trends would partly compensate for the larger standard uncertainty or when only few foraminifera specimens are available. Further, LA, like SIMS, has the potential to gain insight into ontogenetic δ[11]B variations, helping to better understand the biological uptake of boron during chamber formation.





## 5 Conclusions

Microanalytical methods such as SIMS or LA-MC-ICPMS are potentially powerful tools for studying biomineralization processes or possible alternatives to conventional bulk solution analysis of $\delta^{11}B$ in benthic foraminifera, if sample material is limited. For this study, we measured a population of 23 *C. wuellerstorfi* in total using SIMS and femtosecond LA-MC-ICPMS and compared the results with the bulk-solution $\delta^{11}B$, revealing consistent average values among the different techniques. While the medians agree to within ±0.1 ‰, a large intra-shell variability was observed, with up to 6.8 ‰ and 4.5

255 ‰ (2σ) derived from the SIMS and LA methods, respectively. We propose that the larger spread for SIMS, compared to LA, can be attributed to the much smaller volume ($\sim200^{-1}$) of calcite being analyzed in each run, and hence supposedly reflects a larger heterogeneity of $\delta^{11}B$ in the foraminiferal test on a smaller scale. When analytical uncertainties and scale-dependent differences in $\delta^{11}B$ variations are taken into account, the intra-shell variability is likely in the order of ±0.4 and 3 ‰ (2σ) on a 100 and 20 μm scale, respectively.

The $\delta^{11}B$ variability between shells exhibits total ranges of ~3 ‰ for both techniques, suggesting that a number of shells needs to be analyzed for accurate mean $\delta^{11}B$ values. We applied a simple resampling method and conclude that about 7 shells of *C. wuellerstorfi* must be analyzed using LA-MC-ICPMS to obtain an accurate average value to within ±0.5 ‰ (2σ). Hence, we suggest that, based on this high number of required individuals, the bulk solution MC-ICPMS method remains the first choice for analysis of $\delta^{11}B$ in routine paleo-pH studies.

**Data availability**

The boron isotope data collected for this study are available from Table S1 in the supplement.

**Author contribution**

MR, CRB, IH, and JB conceived the study (conceptualization). MR, CRB, and PL carried out measurements, analyzed the data, and performed data statistics (data curation, formal analysis, investigation). AB, KUR, and GS maintained and

270 provided access to analytical instruments at AWI (resources). JB, CRB, and IH raised funding for the French-German project 'B2SeaCarb' (funding acquisition). MR produced the figures for the manuscript (visualization). MR and CRB wrote the first draft of the mansucript (writing – original draft), and all authors interpreted, edited, and reviewed the manuscript (writing – review & editing).

**Competing interests**

The authors declare that they have no conflict of interest.



**Acknowledgements**

This research was carried out in the framework of the joint French/German project 'B2SeaCarb' and was supported by the Deutsche Forschungsgemeinschaft (DFG) grant number BI 432/10-1 to JB and DFG grant number HO 3257/5-1 to IH. On the french side, the project was supported by the French National Research Agency (ANR) grant number ANR-16-CE92-0010 to CRB. CRB thanks N. Bouden (CRPG) for his technical help, and the MARUM GeoB core repository is acknowledged for providing sediment samples.

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





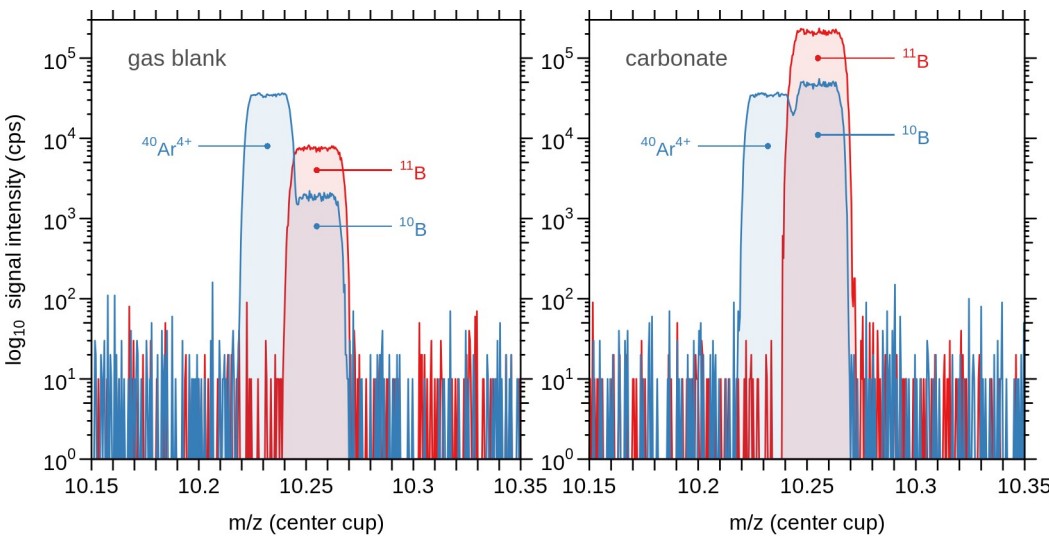

**Figure 1: Mass scans over atomic masses 10 (blue) and 11 (red), centered at ~10.26 amu using Daly detectors. Left: Gas blank (laser off), showing the typical double peak of $^{40}Ar^{4+}$ and $^{10}B$, and the $^{11}B$ peak. Right: Signal of ablated calcium carbonate (laser on). The baseline exhibits only electronic noise from the Daly detectors, but no sign of unresolved interferences on $^{10}B$ as matrix-induced scattered Ca ion. Note that the signal intensity is on a logarithmic scale.**

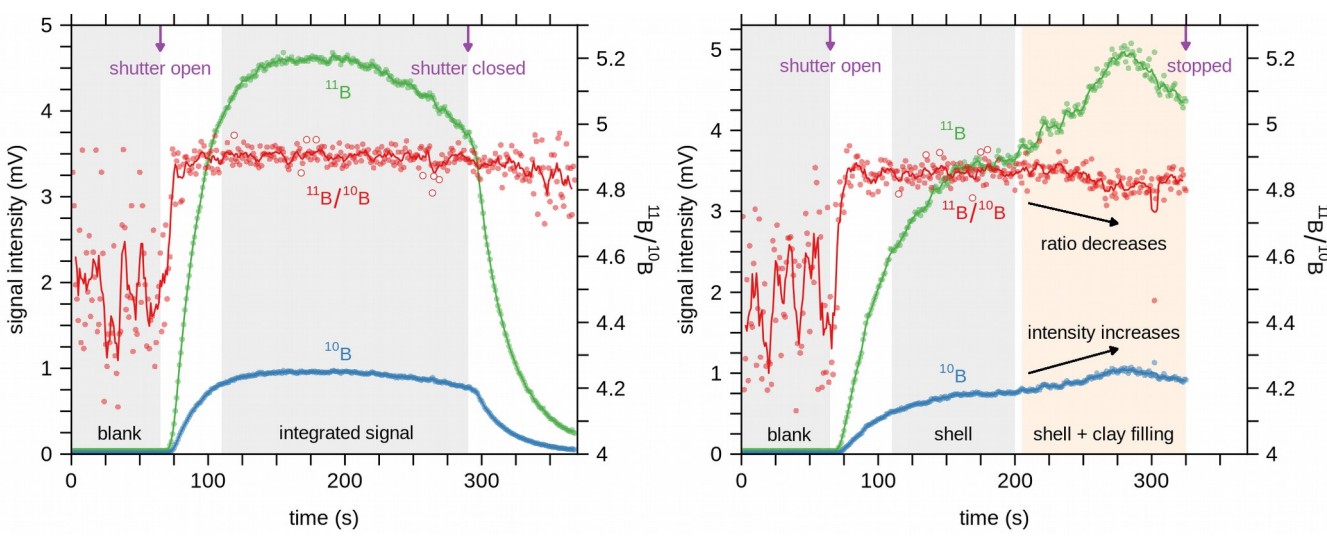

**Figure 2: Left: Typical time-resolved laser ablation analysis for $^{10}B$ and $^{11}B$ of a *C. wuellerstorfi* shell using Daly detectors, preceded by a ~60 s blank measurement. Dots represent 1-s cycles, and lines 5-pt running averages. Open symbols are data that** 290 **are excluded by the 2σ outlier test. Right: Example of a shell that was penetrated by the laser beam, resulting in the ablation of clay infillings.**

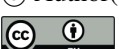



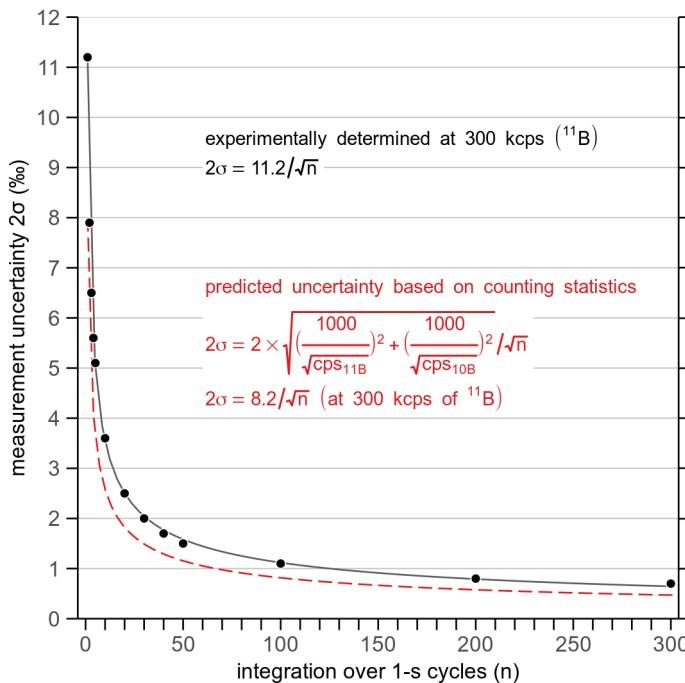

**Figure 3: Measurement uncertainty of $^{11}B/^{10}B$ (2σ) at count rates of 300,000 cps ($^{11}B$) as a function of the laser ablation time. The uncertainty of each boron isotope measurement is calculated based on this relationship (black solid line). A major portion (~70 %) of the measurement uncertainty is related to Poisson-distributed counts (red dashed line).**





**Figure 4: Intra-shell variability of δ¹¹B using SIMS (left panel) and LA-MC-ICPMS (right panel) on selected large individuals of**
*C. wuellerstorfi*. **The residual Δδ¹¹B (difference between single spot and mean δ¹¹B, eq. 2) averaged from all analyzed specimens is**
**shown for each chamber (f is the final chamber, f-1 the penultimate one, and so on). Orange color stands for higher-than-mean**
**and blue for lower-than-mean values. Lighter colors indicate data that are based on only one measurement. The inlet table**
**summarizes the measured intra-shell variability derived from the two techniques.**





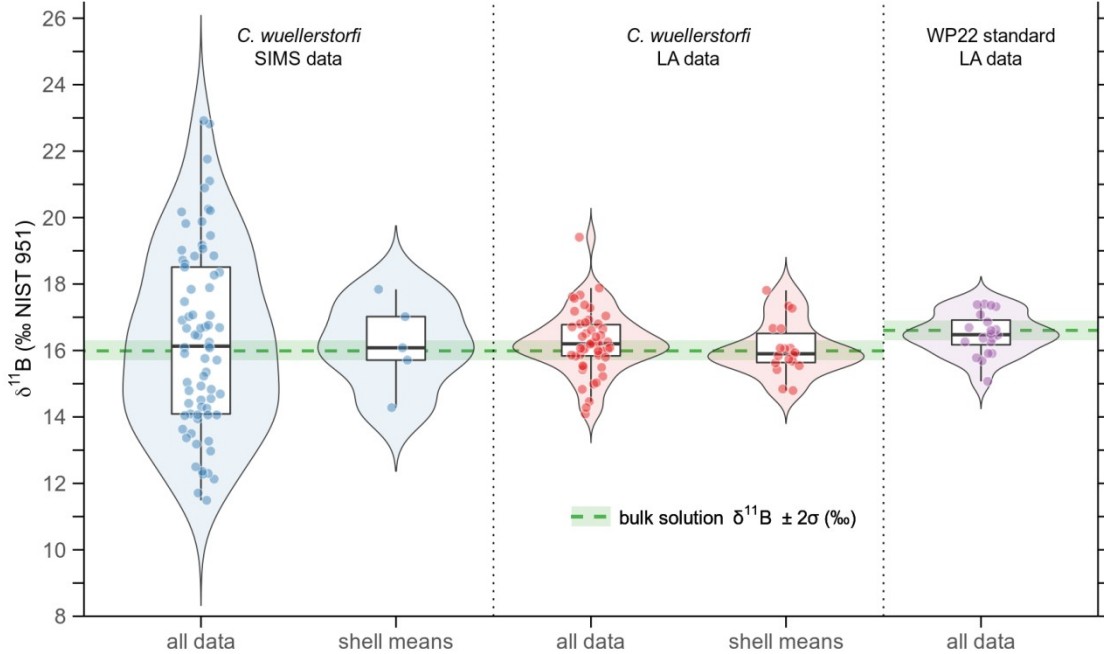

**Figure 5: Violin, box and jitter plots showing the distribution of all single-site δ¹¹B values and single-shell means, both for the SIMS and laser ablation techniques. For comparison, the distribution of δ¹¹B values measured on the in-house reference material WP22 is displayed as well. The green dashed lines and bars represent the bulk solution δ¹¹B ± 2σ values.**

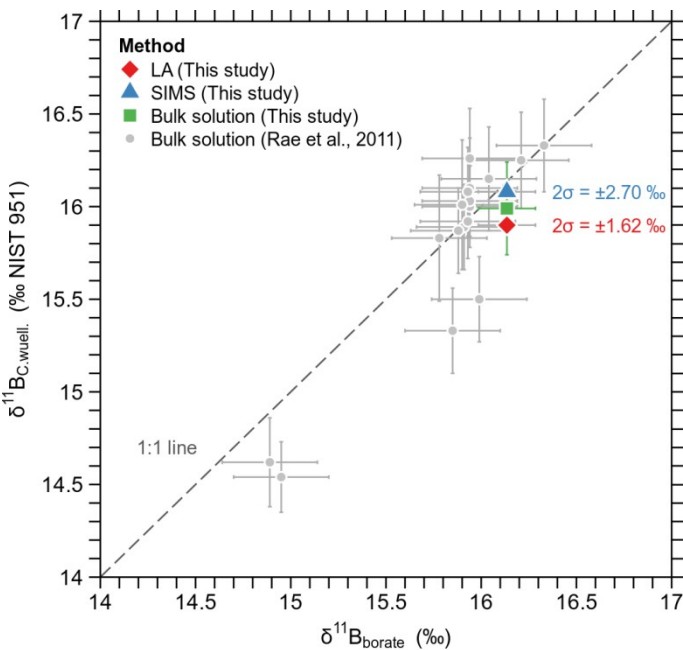

**Figure 6: Median δ¹¹B of Holocene (5.6 ka)** *C. wuellerstorfi* **from GeoB core 1032 (Walvis Ridge, South Atlantic) measured with different techniques, shown along with the core-top calibration from (Rae et al., 2011). Note that the bulk solution analysis of this study was carried out on the same population measured before with laser ablation. Pooled δ¹¹B uncertainties for SIMS (n=5 shells) and LA-MC-ICPMS (n=18 shells) are shown as numbers, as error bars exceed the y-axis scale.**





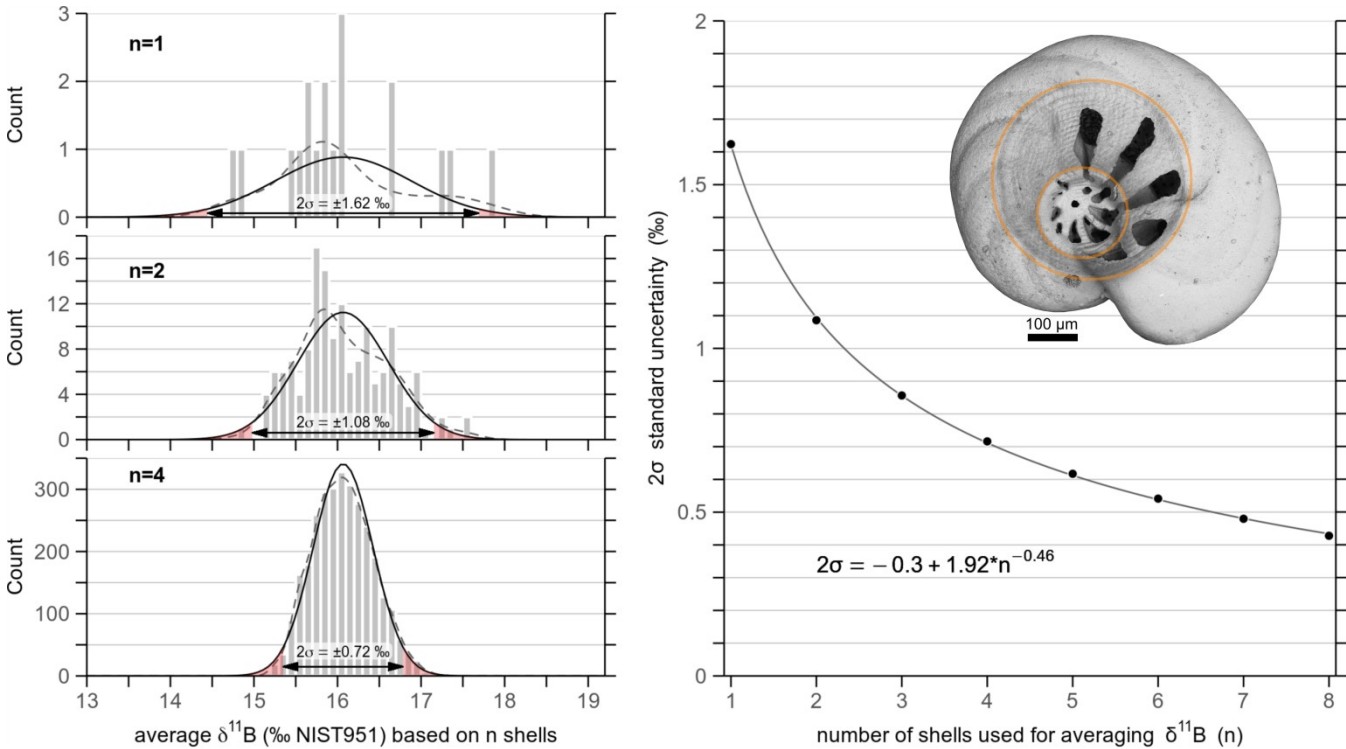

**Figure 7: Simulation of 2σ uncertainty for δ¹¹B using LA-MC-ICPMS in relation to the number of analyzed *C. wuellerstorfi* shells.** The grey bars represent the numbers of mean δ¹¹B values in 0.1 ‰ bins from all possible combinations of n shells out of the original population (n=1). The dashed lines are density curves, and the solid lines normal distribution curves. The more shells are used for analysis, the smaller gets the 2σ uncertainty and the higher the probability to attain the "true" mean value. As an example, the δ¹¹B 2σ uncertainty is ±0.6 ‰ for the average out of 5 shells from the population. The inlet SEM picture shows a specimen measured for "whole-shell" δ¹¹B, typically implemented by a large crater covering a major shell part, and a smaller one on the thicker umbilical knob.