# Peer review of "Technical Note: Single-shell $\delta^{11}$ B analysis of *Cibicidoides wuellerstorfi* using femtosecond laser ablation MC-ICPMS and secondary ion mass spectrometry"

_Biogeosciences, 2020_

## Referee Comment (RC1) · Dennis Mayk (Referee) · 22 Jul 2020

Dear Editor and Authors,

Raitzsch et al. present an interesting and timely manuscript about a comparative study of B isotopes in the benthic foraminifera Cibicidoides wuellerstorfi analysed using LA-MC-ICP-MS and SIMS. Despite the relevance of $\delta$11B as a paleo-pH-proxy, very few studies have been published showing intra and inter foraminfera test (shell) $\delta$11B variability as these analysis have proven to be challenging due to low B concentration and

fragility of foraminifera tests. This study provides an interesting comparison between different heterogeneity levels within and between individual foraminfera which will be of widespread interest and should be published after revision of the issues listed below: Main comments:

—

1. Data processing:

The manuscript lacks a general explanation of how the data were treated after collection.

Fig. 2 shows a typical time-resolved laser ablation profile for a clean and a contaminated (clay filled) foraminifer. In the caption, it is mentioned that some points have been removed from the ablation trend by a 2-sigma outlier test, however in the methods there is no explanation of the data processing involved.

It would be important to mention the general data reduction routine that was employed.

Furthermore, the ablation intensity profiles appear very bulgy and do not present apparent plateaus. Please report how the shell signal was extracted from the rest?

—

2. Sample size estimation:

The estimation of the required sample size to resolve 0.1 pH unit is a very important part of the manuscript but the R function "combn()" used for that purpose lacks a detailed explanation in the manuscript – in addition it is unclear if the presumptions made in the manuscript are correct or lead to an underestimation of the required sample size.

In detail:

On line 237 it is reported that the sample size simulation is based on the assumption that the entire population (P) consists of the 18 shells analysed. Although this holds

true for this particular study it is not a representation of the actual (true) population size which is what future studies would be interested in to estimate required sample sizes. In other words, the presumption of P = 18 holds only true within this study but has no real world application. Instead it should be discussed what population sizes are realistic within similar pH-environments and simulations should be based on these.

Furthermore, it is not clear how the simulation are carried out using the 18 shells as they have not been measured in the same way according to the Supplemental Material. The "large crater" was analysed on 16 shells and the "umbilical knob" was also analysed on 16 shells suggesting that for the simulation using 18 shells two different measurement "types" were merged which further complicates its validity. It would be more informative to separate the two and report required sample sizes based on measurement type i.e., for measurements on the "umbilical knob" and for "large crater" measurements.

Considering the sample size of 16 or 18 it appears to be useful to consider the use of a conventional sample size estimation approach in comparison to a resampling approach as drawing from the same small population may result in errors. In the figure below, the estimated sample sizes required for e = 0.5 (2SD) and 1-$\alpha$ = 0.05 in relation to the population size is given as estimated by the R function "sample.size.mean()" (https://CRAN.R-project.org/package=samplingbook) for both measurement "types". Given an overall population size of e.g., 500 specimens in the same pH-environment, it would require n = 87 specimens if the "umbilical knob" was measured and n = 40 if the "large crater" was measured (based on the variability observed in this study) to achieve the desired significance level. Even if the population size consisted of only 16 individuals, the estimated sample size would be n = 16 and n = 14, respectively and thus twice as large as reported in the manuscript.

—

Minor comments:

This is a non-comprehensive list of minor issues.

Line 30: Consider removing the last clause of the abstract. "Vital effect" is a loaded term and since it is not further discussed (Line 185) of little value for this manuscript.

Line 35: Space missing between 27.2 and $\pm 0.6$ ‰

Line 57: Comma missing after "Also"

Line 69: Considering that this study looked at a total of 23 specimens the term "tens of specimens" seems excessive, better report the actual number of individuals.

Line 179: Why was a non-parametric test used? Please specify what data the test was used on? Please report the Wilcoxon-Mann-Whitney test summary i.e., (W = XXX, p <0.001)

Line 184: Space missing between "large-scale" and "suggesting"

Line 186: "Somewhat" not useful, report how much $\delta 11B$ was elevated in the umbilical knob

Line 197: a total of 18 shells "were" used

—

Sincerely,

Dennis Mayk

Please also note the supplement to this comment:
https://www.biogeosciences-discuss.net/bg-2020-269/bg-2020-269-RC1-supplement.pdf

[Figure]

**Fig. 1.**

---

## Referee Comment (RC2) · Lubos Polerecky (Referee) · 5 Aug 2020

Raitzsch et al. provide a detailed comparison between SIMS and LA-MC-ICPMS measurements of delta-11B in individual shells of benthic foraminifera. They show that intra-shell and inter-shell variability is significantly lower for the LA-based technique compared with SIMS, which they attribute to the larger volume sampled by the LA-based technique. Importantly, they show that both techniques yield very similar "average" values to those obtained by the traditional bulk measurements based on dissolved specimens. They conclude that the traditional bulk-based analysis is still the preferred

approach for paleo applications, but demonstrate clearly the advantages and limits of the microanalytical techniques.

The manuscript is well written and clearly organized. Also the figures are clear and of excellent quality.

I only have a few minor comments and questions. I recommend the manuscript for publication after these minor issues have been resolved by the authors.

Technical comments/questions/suggestions:

l.69: Please formulate more clearly the *aim* of the study. 'What' do you want to achieve, and especially 'why'?

l.154-155: Please clarify how this variability was calculated. Since 2*sigma is used, it may be confused with 2*sigma of the individual measurement's precision. And since per-mil units are used, it may be confused with the coefficient of variation (which is in per-cent). To avoid confusion, best would be to clarify in one sentence that 2*sigma here actually corresponds to 2*SD of n individual measurements (if I understand it correctly). Or is it 2*SE (standard error)?

l.157: unclear why such inistinguishability should affect variation in measured data. Please explain, or provide an alternative explanation.

l.168-169: Please clarify how this was derived/deduced. Intuitively it is expected that variability in measurements is lower if larger volume is sampled. But it is unclear how you arrived to those values (e.g. ∼0.3 permil).

l.176/fig.4: Please clarify representation of the data in polar plots in Fig. 4. I understand that the "phi" coordinate corresponds to the chamber, but it took me a while to figure out that the r-coordinate (scale -7 to 3) corresponds to Delta-delta11B.

Also I am wondering whether it would be more beneficial/transparent to show each Delta-delta-11B datapoint rather than average Delta-delta-11B deviations derived from

measurements of multiple specimens. Averages may be misleading, as we know. Did you test whether the decreasing trend between f and f-5 is significant, or you can only state "the deviation tends to decrease"?

l.200: I am wondering why the authors report median instead of, for example, the mean? If it leads to a different mean in comparison to the bulk-based analysis, it should be discussed why such a difference exists. In any case, I think it needs to be clarified why median was used. Similar on l.254.

l.211: yes, I agree, but it would be useful to expand this argument towards the *source* of this variability (e.g., shell-to-shell differences in the intra-shell heterogeneity?).

Figure 1: It is rather confusing to see signals for 10B and 11B centred on the same mass (10.25). Is it really so? And why? I am not familiar with the Daly detector principle.

Figure 3: Please verify the expression for 2*sigma in the graph (in red). First, the factor 1000 does not make sense if cps is in counts per second (perhaps it does if it is in kilo-counts per second). Second, if I substitute 300,000 and 300,000/4.9 for 11B and 10B, I get a factor 8.8, not 8.2. In my opinion, the formula should read as 2*sqrt(1/counts(11B)+1/counts(10B)), where counts(11B)=cps(11B)*1s*n and similarly for 10B. This is a formula for the Poisson error of 11B/10B based on counting statistics. In this formula the factor is then 0.00887 at B11=300,000. Please verify cps vs. kcps.

Editorial suggestions:

l.24: unclear why the word "presumably" is used in the abstract. It would help if the sentence is reworded to clarify what is certain and what is not (i.e., what is estimated).

l.39: would be useful to cite few examples of such studies.

l.104: perhaps it should read "45 cm4 *and* ablated"?

l.126: remove "and" before "that"

In caption to Fig. 4, it should read "inset", not "inlet". Similar on l.195.

l.184: remove "large-scale"

l.279: "French" - uppercase F.

---

## Short Comment (SC1) · 10 Aug 2020

Very nice work! It will be a great contribution to the community.

Line 109: Should be 11B/10B?

Line: 184 Delete "large-scale"

Lines: 235-243: The readers may want to know more detail on the simulation.

Figure 7: It is interesting attempt, but I could not understand how it is simulated. If n =

4, count should be 52? (13*4) Why so much count is obtained in this simulation?

---

## Author Comment (AC1) · 9 Sep 2020

AC: We thank Dennis Mayk for his constructive and thorough review of our manuscript and will address all his remarks and suggestions, which are listed below:

Raitzsch et al. present an interesting and timely manuscript about a comparative study of B isotopes in the benthic foraminifera Cibicidoides wuellerstorfi analysed using LA-MC-ICP-MS and SIMS. Despite the relevance of $\delta$11B as a paleo-pH-proxy, very few studies have been published showing intra and inter foraminfera test (shell) $\delta$11B variability as these analysis have proven to be challenging due to low B concentration and fragility of foraminifera tests. This study provides an interesting comparison between different heterogeneity levels within and between individual foraminfera which will be of widespread interest and should be published after revision of the issues listed below:

Main comments: 1. Data processing: The manuscript lacks a general explanation of how the data were treated after collection. Fig. 2 shows a typical time-resolved laser ablation profile for a clean and a contaminated (clay filled) foraminifer. In the caption, it is mentioned that some points have been removed from the ablation trend by a 2-sigma outlier test, however in the methods there is no explanation of the data processing involved. It would be important to mention the general data reduction routine that was employed.

AC: Right, the explanation of the data reduction routine is obviously too sparse (l. 119-120), and will be replaced by a more detailed version.

Furthermore, the ablation intensity profiles appear very bulgy and do not present apparent plateaus. Please report how the shell signal was extracted from the rest?

AC: The reason for the bulgy shape of the signal intensities in Fig. 2 is probably related to changing ablation efficiency for some samples, i.e. a more efficient ablation of material from a surface progressively getting rougher after a couple of "helix turns". As we always attempted to match the signal, i.e. to gain the same signal intensities between sample and standard, we often had to increase the ablation frequency to enhance ablation at the beginning of a measurement. Conversely, after some time, we often had to decrease the frequency, when ablation was liable to become too strong, resulting in a bulgy profile as shown in Fig. 2. For integration of the shell signal, we chose an interval, where the signal ratio clearly showed a smooth plateau (see Fig. 2), which will be better explained in the revised manuscript.

2. Sample size estimation: The estimation of the required sample size to resolve 0.1 pH unit is a very important part of the manuscript but the R function "combn()" used for

that purpose lacks a detailed explanation in the manuscript – in addition it is unclear if the presumptions made in the manuscript are correct or lead to an underestimation of the required sample size. In detail: On line 237 it is reported that the sample size simulation is based on the assumption that the entire population (P) consists of the 18 shells analysed. Although this holds true for this particular study it is not a representation of the actual (true) population size which is what future studies would be interested in to estimate required sample sizes. In other words, the presumption of P =18 holds only true within this study but has no real world application. Instead it should be discussed what population sizes are realistic within similar pH-environments and simulations should be based on these. Furthermore, it is not clear how the simulation are carried out using the 18 shells as they have not been measured in the same way according to the Supplemental Material. The "large crater" was analysed on 16 shells and the "umbilical knob" was also analysed on 16 shells suggesting that for the simulation using 18 shells two different measurement "types" were merged which further complicates its validity. It would be more informative to separate the two and report required sample sizes based on measurement type i.e., for measurements on the "umbilical knob" and for "large crater" measurements.

AC: This is a good point raised by Dennis, as only the 14 and 16 individuals were analyzed using the "umbilical knob" and "large crater", respectively. Hence, for the analysis of "sample size requirement" we will exclude the 2 individuals analyzed for inter-chamber variability, and only examine the separate variabilities based on "knob" and "large crater" analyses, as well as on the variability where both analysis types are averaged. Accordingly, we will update Fig. 7 showing the results.

Considering the sample size of 16 or 18 it appears to be useful to consider the use of a conventional sample size estimation approach in comparison to a resampling approach as drawing from the same small population may result in errors. In the figure below, the estimated sample sizes required for e = 0.5 (2SD) and 1-$\alpha$ = 0.05 in relation to the population size is given as estimated by the R function "sample.size.mean()"

(https://CRAN.R-project.org/package=samplingbook) for both measurement "types". Given an overall population size of e.g., 500 specimens in the same pH-environment, it would require n = 87 specimens if the "umbilical knob" was measured and n = 40 if the "large crater" was measured (based on the variability observed in this study) to achieve the desired significance level. Even if the population size consisted of only 16 individuals, the estimated sample size would be n = 16 and n = 14, respectively and thus twice as large as reported in the manuscript.

AC: Also this is a very important point, which made us to rethink about this issue. Firstly, it is right that our approach is not the most appropriate resampling method, as the entire data population only consists of the 18 measured individuals. The R function "combn()" searches for all possible combinations within this population and hence does not apply replacement of a sample, i.e. it does not resample one foram multiple times for generating one subsample. This ultimately results in an underestimation of the uncertainty as a function of measured individuals, which was also correctly pointed out by Dennis. However, we think that the resampling approach proposed by Dennis (R function "sample.size.mean()") is not the most appropriate neither. That is because this function assumes that the value of a measurand from the entire population may be approached by measuring a random subsample, the size of which is dependent on the population size and the target uncertainty. In other words, this function allows for determining the required subsample size in order to gain the "true" average value of the entire population to within a quoted uncertainty, but it does not reflect whether the average value accurately records the influencing variable, in our case pH that influences $\delta 11B$. The output plot provided by Dennis implies that a few individuals are sufficient to gain an accurate value, if the population is small, but this is not true as each specimen has a large uncertainty in terms of the closeness of the agreement between the measured $\delta 11B$ and the influencing pH. Consequently, the relationship between the "accuracy" and number of analyzed individuals must be independent of the population size.

[Figure]

Based on these thoughts, we will apply a different method, but which partly goes the same direction as the resampling approach suggested by Dennis. In the revised manuscript, we will use a Monte Carlo simulation, where a large artificial population (n=10,000) is created by randomly generating $\delta$11B values around the "true" $\delta$11B value within the determined individual uncertainty of $\pm0.84$ and $\pm1.38$ ‰ (SD) for "large craters" and "umbilical knobs", respectively. From this population, we will randomly re-sample and average N values to determine the 2SD uncertainty (= the potential error of $\delta$11B) as a function of N analyzed individuals. We think that this approach is the most appropriate one, as it is independent of the population size and does not underestimate the uncertainty, as does our initial approach.

Minor comments: This is a non-comprehensive list of minor issues Line 30: Consider removing the last clause of the abstract. "Vital effect" is a loaded term and since it is not further discussed (Line 185) of little value for this manuscript.

AC: Correct, will be removed.

Line 35: Space missing between 27.2 and $\pm0.6$ ‰

AC: OK.

Line 57: Comma missing after "Also"

AC: OK.

Line 69: Considering that this study looked at a total of 23 specimens the term "tens of specimens" seems excessive, better report the actual number of individuals.

AC: Right, will be changed.

Line 179: Why was a non-parametric test used? Please specify what data the test was used on? Please report the Wilcoxon-Mann-Whitney test summary i.e., (W = XXX, p <0.001)

AC: We used a non-parametric test because it does not imply defined probability distributions a priori, but is open to the model structure. The WMW uses randomly selected values X and Y from two populations, and tests the null hypothesis whether the probability that X>Y is equal to the probability that Y>X. However, I am very happy that Dennis made this a subject of discussion, since I walked right into a trap when testing the null hypotheses. Because of the few datapoints for each chamber, I applied the statistical test on Monte-Carlo simulated d11B values (n=10) around quoted uncertainties, yielding p values smaller than 0.05, meaning that the differences in d11B between chambers f-1 and f-5 are statistically significant at a 95 % SL. This artificially increased population size, however, led to a biased uncertainty estimation, which was also subject to papers in mathematical journals (e.g. Lin et al. (2013), Too Big to Fail: Large Samples and the p-Value Problem, http://dx.doi.org/10.1287/isre.2013.0480). If just the original data are taken into account, both the WMW and Welch t-test yield p values $\sim$0.07, and hence the d11B differences between the chambers are not statistically different at a 95 % SL, based on the small datasets of this study. I have to apologize for this incautious and naive application of statistical tests on our data. This will be clarified in the revised manuscript.

Line 184: Space missing between "large-scale" and "suggesting"

AC: OK

Line 186: "Somewhat" not useful, report how much $\delta11B$ was elevated in the umbilical knob

AC: OK, will be changed.

Line 197: a total of 18 shells "were" used

AC: OK, thanks.
* * *

---

## Author Comment (AC2) · 9 Sep 2020

AC: Lubos Polerecky's very helpful comments and suggestions, particularly from his analytical view, on our manuscript are very much appreciated and will all be addressed in the revised manuscript (listed below).

Raitzsch et al. provide a detailed comparison between SIMS and LA-MC-ICPMS measurements of delta-11B in individual shells of benthic foraminifera. They show that intra-shell and inter-shell variability is significantly lower for the LA-based technique

compared with SIMS, which they attribute to the larger volume sampled by the LA-based technique. Importantly, they show that both techniques yield very similar "average" values to those obtained by the traditional bulk measurements based on dissolved specimens. They conclude that the traditional bulk-based analysis is still the preferred approach for paleo applications, but demonstrate clearly the advantages and limits of the microanalytical techniques. The manuscript is well written and clearly organized. Also the figures are clear and of excellent quality. I only have a few minor comments and questions. I recommend the manuscript for publication after these minor issues have been resolved by the authors.

Technical comments/questions/suggestions: l.69: Please formulate more clearly the *aim* of the study. 'What' do you want to achieve, and especially 'why'?

AC: That's right, the aim of the study is not clearly enough explained, but just vaguely outlined between l. 38-47. We will emphasize this in the revised version.

l.154-155: Please clarify how this variability was calculated. Since 2*sigma is used, it may be confused with 2*sigma of the individual measurement's precision. And since permil units are used, it may be confused with the coefficient of variation (which is in percent). To avoid confusion, best would be to clarify in one sentence that 2*sigma here actually corresponds to 2*SD of n individual measurements (if I understand it correctly). Or is it 2*SE (standard error)?

AC: Yes, the reported 2*sigma variability is the 2-fold standard deviation, derived from the individual measurements. This is not to be confused with the measurement uncertainty (=precision), which is dependent on the ablation time and is also given as 2*sigma (=2*SD). So we use sigma as a statistical expression just to clarify that the SD is reported, and not the SE. We will explain it in more detail in the revised manuscript.

l.157: unclear why such inistinguishability should affect variation in measured data. Please explain, or provide an alternative explanation.

AC: Right, this part is quite confusing and probably also not reflecting the truth. It is true that we observed the largest variability in the knob area that might be attributed to a signal mixture from multiple juvenile chambers, but it may also be due to the higher number of measurements compared to the other chambers. In addition, there were also similarly large variabilities found in chambers f-8 and f-9. So we have to revise (or delete) the according statement. Thanks for hinting at this inconsistency.

l.168-169: Please clarify how this was derived/deduced. Intuitively it is expected that variability in measurements is lower if larger volume is sampled. But it is unclear how you arrived to those values (e.g. âĹij0.3 permil).

AC: Thanks, we missed to describe how we calculated this. We simply applied the following function to estimate the variability reduction: $u(V2) = u(V1)/sqr(V2/V1)$, where $u(V1)$ is the variability for a quoted volume, and $V1$ and $V2$ represent the two different volumes that are compared. We will add this information.

l.176/fig.4: Please clarify representation of the data in polar plots in Fig. 4. I understand that the "phi" coordinate corresponds to the chamber, but it took me a while to figure out that the r-coordinate (scale -7 to 3) corresponds to Delta-delta11B. Also I am wondering whether it would be more beneficial/transparent to show each Delta-delta-11B datapoint rather than average Delta-delta-11B deviations derived from measurements of multiple specimens. Averages may be misleading, as we know.

AC: I agree, the so-called coxcomb chart seems to be difficult to read at the beginning, but once it's understood, it is a nice way of presenting such data. In the updated figure, it will be clearer that red represents positive and blue negative Delta-delta values. I have tried a couple graph types, also plotting all individual datapoints, as Lubos suggested, but this resulted in a quite confusing graph due to the large number of measurements. For our aim to eventually observe trends, plotting averaged deviations seems to be the catchiest way.

Did you test whether the decreasing trend between f and f-5 is significant, or you can

only state "the deviation tends to decrease"?

AC: Yes, we applied the Wilcoxon-Mann-Whitney approach to test the significance of d11B differences between chambers (l. 178 ff). Also, Dennis Mayk made this a subject of discussion, and I realized that I have walked right into a trap when testing the null hypotheses. Because of the few datapoints for each chamber, I applied the statistical test on Monte-Carlo simulated d11B values (n=10) around quoted uncertainties, yielding p values smaller than 0.05, meaning that the differences in d11B between chambers f-1 and f-5 are statistically significant at a 95 % SL. This artificially increased population size, however, led to a biased uncertainty estimation, which was also subject to papers in mathematical journals (e.g. Lin et al. (2013), Too Big to Fail: Large Samples and the p-Value Problem, http://dx.doi.org/10.1287/isre.2013.0480). If just the original data are taken into account, both the WMW and Welch t-test yield p values ∼0.07, and hence the d11B differences between the chambers are not statistically different at a 95 % SL, based on the small datasets of this study. I have to apologize for this incautious and naive application of statistical tests on our data. This will be corrected in the revised manuscript.

l.200: I am wondering why the authors report median instead of, for example, the mean? If it leads to a different mean in comparison to the bulk-based analysis, it should be discussed why such a difference exists. In any case, I think it needs to be clarified why median was used. Similar on l.254.

AC: The differences between medians and means are small, e.g. the SIMS median is 16.08 ‰ and the mean is 16.19 ‰ while the LA median is 15.91 ‰ and the mean is 16.17 ‰.However, the median is less sensitive against outliers than the mean and also represents the average value of a non-Gaussian distribution. The median is also equivalent to the average shown by the violin/box plots in Fig. 5. We will insert a small sentence why we chose the median instead of the mean.

l.211: yes, I agree, but it would be useful to expand this argument towards the *source*

of this variability (e.g., shell-to-shell differences in the intra-shell heterogeneity?).

AC: We think this goes slightly beyond the scope of this study, and is difficult to answer, based on our database. Maybe the range of isotopic compositions of the trigonal BO3 hosted in the calcite lattice is very large (not yet examined on the molecular scale), and hence better resolved the smaller the scale of the analytical technique is.

Figure 1: It is rather confusing to see signals for 10B and 11B centered on the same mass (10.25). Is it really so? And why? I am not familiar with the Daly detector principle.

AC: On a multicollector ICP-MS, the Quad lenses are tuned in a way that the peaks of different isotopes (here 10B and 11B) coincide, i.e. the incoming ions hit the respective detectors simultaneously, where the one for high mass (11B) is on either and the low mass (10B) on the other side of the Center cup. Once the peaks coincide and the peak center is set, the information on the position of the peak center is read by the Center cup. Of course, 11B is measured on 11.009 amu and 10B on 10.013 amu, but the position of the peak center is in relation to the Center cup, and might slightly change on a daily basis, depending on the tuning parameters. We will slightly modify the figure caption to make it clearer.

Figure 3: Please verify the expression for 2*sigma in the graph (in red). First, the factor 1000 does not make sense if cps is in counts per second (perhaps it does if it is in kilo-counts per second). Second, if I substitute 300,000 and 300,000/4.9 for 11B and 10B, I get a factor 8.8, not 8.2. In my opinion, the formula should read as 2*sqrt(1/counts(11B)+1/counts(10B)), where counts(11B)=cps(11B)*1s*n and similarly for 10B. This is a formula for the Poisson error of 11B/10B based on counting statistics. In this formula the factor is then 0.00887 at B11=300,000. Please verify cps vs. kcps.

AC: The factor 1000 is because the boron isotopic composition is given in permil (see eq. 1 in the main text), so it's expressing the relative difference from the standard value. The measurement uncertainty (i.e. the internal precision) must hence be multiplied

with 1000 to have the number in permil as well. We agree that the formula we provided is quite cumbersome, but it is mathematically correct. Based on the simpler formula provided by Lubos, we slightly modified it to more easily enter the number of cycles (n), and multiply if with the factor 1000 to obtain the result in permil expression. The final formula is now 2*sqrt(1/cps(11B)+1/cps(10B)) / sqrt(n)*1000, which will replace the current one in the revised figure. Concerning the obtained factor at a countrate of 300,000 cps for 11B, the ratio between 11B and 10B is in nature in the order of 4, and not 4.9 (11B=80.1 %, 10B=19.9 %). Therefore, if 11B is measured at 300,000 cps, 10B is recorded at approximately 75,000 cps, thus giving a factor of 8.8 using the formula above. We will add the expected countrate on 10B in the revised figure.

Editorial suggestions: l.24: unclear why the word "presumably" is used in the abstract. It would help if the sentence is reworded to clarify what is certain and what is not (i.e., what is estimated).

AC: Right, "presumably" is a too careful term. We will replace it with "estimated to be".

l.39: would be useful to cite few examples of such studies.

AC: These are the same references as in lines 32-33, but we will list them here as well.

l.104: perhaps it should read "45 cm3 *and* ablated"?

AC: Thanks, will be corrected.

l.126: remove "and" before "that"

AC: OK.

In caption to Fig. 4, it should read "inset", not "inlet". Similar on l.195.

AC: Right, slip of mine.

l.184: remove "large-scale"

AC: Thanks for that, was a leftover from a former sentence.

l.279: "French" - uppercase F.

AC: Will be corrected.
* * *

---

## Author Comment (AC3) · 9 Sep 2020

AC: We appreciate the interactive comment of Kaoru Kubota and his careful reading of our manuscript, and will address his comments below:

Very nice work! It will be a great contribution to the community.

AC: Thank you.

Line 109: Should be 11B/10B?

AC: Well spotted! Will be corrected.

Line: 184 Delete "large-scale"

AC: Thanks, will be deleted.

Lines: 235-243: The readers may want to know more detail on the simulation.

Figure 7: It is interesting attempt, but I could not understand how it is simulated. If n = 4, count should be 52? (13*4) Why so much count is obtained in this simulation?

AC: Yes, it is an interesting approach, but it is not the most appropriate one, as also Dennis Mayk suggested, because it results in an underestimation of the uncertainties. We will thus apply a Monte Carlo approach in the revised manuscript, which is similar, but with correct uncertainty estimations.

However, just for information on the combn() function. It uses an input population (e.g., A, B, C, D) and calculates the averages for all possible combinations among this population (k), for a given number of subsamples (n). For instance, for n=2 it calculates the averages AB, AC, AD, BC, BD, CD, so we get 6 possible combinations. To calculate the number of possible combinations (N) for any k and n, the binomial coefficient is used: $N=k!/(n!*(k-n)!)$. The possible combinations of 4 samples out of a total of 18, as in Kaoru's example, thus amount to 3060.

---

## Author Response (AR1)

**Author's response**

I    Point-by-point reply to the reviewers' comments

II    Marked-up manuscript version

**I    Point-by-point reply to the reviewers' comments**

**Review No 1 (Dennis Mayk)**

AC: We thank Dennis Mayk for his constructive and thorough review of our manuscript and addressed all his remarks and suggestions, which are listed below:

Raitzsch et al. present an interesting and timely manuscript about a comparative study of B isotopes in the benthic foraminifera Cibicidoides wuellerstorfi analysed using LA-MC-ICP-MS and SIMS. Despite the relevance of δ11B as a paleo-pH-proxy, very few studies have been published showing intra and inter foraminfera test (shell) δ11B variability as these analysis have proven to be challenging due to low B concentration and fragility of foraminifera tests. This study provides an interesting comparison between different heterogeneity levels within and between individual foraminfera which will be of widespread interest and should be published after revision of the issues listed below:

Main comments:
1. Data processing:
The manuscript lacks a general explanation of how the data were treated after collection. Fig. 2 shows a typical time-resolved laser ablation profile for a clean and a contaminated (clay filled) foraminifer.
In the caption, it is mentioned that some points have been removed from the ablation trend by a 2-sigma outlier test, however in the methods there is no explanation of the data processing involved. It would be important to mention the general data reduction routine that was employed.

AC: Right, the explanation of the data reduction routine was obviously too sparse, and is now complemented by more details (l. 127-132).

Furthermore, the ablation intensity profiles appear very bulgy and do not present apparent plateaus. Please report how the shell signal was extracted from the rest?

AC: The reason for the bulgy shape of the signal intensities in Fig. 2 is probably related to changing ablation efficiency for some samples, i.e. a more efficient ablation of material from a surface progressively getting rougher after a couple of "helix turns". As we always attempted to match the signal, i.e. to gain the same signal intensities between sample and standard, we often had to increase the ablation frequency to enhance ablation at the beginning of a measurement. Conversely, after some time, we often had to decrease the frequency, when ablation was liable to become too strong, resulting in a bulgy profile as shown in Fig. 2. For integration of the shell signal, we chose an interval, where the signal ratio clearly showed a smooth plateau (see Fig. 2), which is now better explained in the revised manuscript (l. 120-126).

2. Sample size estimation:

The estimation of the required sample size to resolve 0.1 pH unit is a very important part of the manuscript but the R function "combn()" used for that purpose lacks a detailed explanation in the manuscript – in addition it is unclear if the presumptions made in the manuscript are correct or lead to an underestimation of the required sample size.

In detail:

On line 237 it is reported that the sample size simulation is based on the assumption that the entire population (P) consists of the 18 shells analysed. Although this holds true for this particular study it is not a representation of the actual (true) population size which is what future studies would be interested in to estimate required sample sizes. In other words, the presumption of P =18 holds only true within this study but has no real world application. Instead it should be discussed what population sizes are realistic within similar pH-environments and simulations should be based on these.

Furthermore, it is not clear how the simulation are carried out using the 18 shells as they have not been measured in the same way according to the Supplemental Material. The "large crater" was analysed on 16 shells and the "umbilical knob" was also analysed on 16 shells suggesting that for the simulation using 18 shells two different measurement "types" were merged which further complicates its validity. It would be more informative to separate the two and report required sample sizes based on measurement type i.e., for measurements on the "umbilical knob" and for "large crater" measurements.

AC: This is a good point raised by Dennis, as only the 14 and 16 individuals were analyzed using the "umbilical knob" and "large crater", respectively. Hence, for the analysis of "sample size requirement" we have excluded the 2 individuals analyzed for inter-chamber variability, and only examine the separate variabilities based on "knob" and "large crater" analyses. This is explained in the main text (l. 246-249). Accordingly, we also modfied Fig. 7 showing the results for each "measurement type".

Considering the sample size of 16 or 18 it appears to be useful to consider the use of a conventional sample size estimation approach in comparison to a resampling approach as drawing from the same small population may result in errors. In the figure below, the estimated sample sizes required for e = 0.5 (2SD) and 1-α = 0.05 in relation to the population size is given as estimated by the R function "sample.size.mean()" (https://CRAN.R-project.org/package=samplingbook) for both measurement "types". Given an overall population size of e.g., 500 specimens in the same pH-environment, it would require n = 87 specimens if the "umbilical knob" was measured and n = 40 if the "large crater" was measured (based on the variability observed in this study) to achieve the desired significance level. Even if the population size consisted of only 16 individuals, the estimated sample size would be n = 16 and n = 14, respectively and thus twice as large as reported in the manuscript.

AC: Also this is a very important point, which made us to rethink about this issue. Firstly, it is right that our approach is not the most appropriate resampling method, as the entire data population only consists of the 18 measured individuals. The R function "combn()" searches for all possible

combinations within this population and hence does not apply replacement of a sample, i.e. it does not resample one foram multiple times for generating one subsample. This ultimately results in an underestimation of the uncertainty as a function of measured individuals, which was also correctly pointed out by Dennis.

However, we think that the resampling approach proposed by Dennis (R function "sample.size.mean()") is not the most appropriate neither. That is because this function assumes that the value of a measurand from the entire population may be approached by measuring a random subsample, the size of which is dependent on the population size and the target uncertainty. In other words, this function allows for determining the required subsample size in order to gain the "true" average value of the entire population to within a quoted uncertainty, but it does not reflect whether the average value accurately records the influencing variable, in our case pH that influences $\delta^{11}B$. The output plot provided by Dennis implies that a few individuals are sufficient to gain an accurate value, if the population is small, but this is not true as each specimen has a large uncertainty in terms of the closeness of the agreement between the measured $\delta^{11}B$ and the influencing pH. Consequently, the relationship between the "accuracy" and number of analyzed individuals must be independent of the population size.

Based on these thoughts, we modified our method, but which partly goes the same direction as the resampling approach suggested by Dennis. In the revised manuscript, we generated Monte Carlo simulations, where a large artificial population (n=10,000) is created by randomly generating $\delta^{11}B$ values around the "true" $\delta^{11}B$ value within the determined individual uncertainty of ±0.84 and ±1.38 ‰ (SD) for "large craters" and "umbilical knobs", respectively. On these populations, we applied the R function "combn()" for randomly resampled $\delta^{11}B$ values to determine the 2SD uncertainty (= the potential error of $\delta^{11}B$) as a function of n (1 to 16) analyzed individuals. Interestingly, we come to the same required sample sizes as Dennis with his approach for the quoted uncertainty of ±0.5 permil, but is more realistic for fewer shells, as it is independent of the original population size. We have partly rewritten/complemented the text (l. 246-258) and redrawn Fig. 7, which is now without histograms.

Minor comments:
This is a non-comprehensive list of minor issues
Line 30: Consider removing the last clause of the abstract. "Vital effect" is a loaded term and since it is not further discussed (Line 185) of little value for this manuscript.

AC: Correct, removed (l. 29).

Line 35: Space missing between 27.2 and ±0.6 ‰

AC: Corrected (l. 34).

Line 57: Comma missing after "Also"

AC: Corrected (l. 56).

Line 69: Considering that this study looked at a total of 23 specimens the term "tens of specimens" seems excessive, better report the actual number of individuals.

AC: Right, changed (l. 68).

Line 179: Why was a non-parametric test used? Please specify what data the test was used on? Please report the Wilcoxon-Mann-Whitney test summary i.e., (W = XXX, p <0.001)

AC: We used a non-parametric test because it does not imply defined probability distributions a priori, but is open to the model structure. The WMW uses randomly selected values X and Y from two populations, and tests the null hypothesis whether the probability that X>Y is equal to the probability that Y>X. However, I am very happy that Dennis made this a subject of discussion, since I walked right into a trap when testing the null hypotheses. Because of the few datapoints for each chamber, I applied the statistical test on Monte-Carlo simulated d11B values (n=10) around quoted uncertainties, yielding p values smaller than 0.05, meaning that the differences in d11B between chambers f-1 and f-5 are statistically significant at a 95 % SL. This artificially increased population size, however, led to a biased uncertainty estimation, which was also subject to papers in mathematical journals (e.g. Lin et al. (2013), Too Big to Fail: Large Samples and the p-Value Problem, http://dx.doi.org/10.1287/isre.2013.0480). If just the original data are taken into account, both the WMW and Welch t-test yield p values ~0.07, and hence the d11B differences between the chambers are not statistically different at a 95 % SL, based on the small datasets of this study. I have to apologize for this incautios and naive application of statistical tests on our data. The text is corrected in the revised manuscript (l. 186-188).

Line 184: Space missing between "large-scale" and "suggesting"

AC: "large-scale" was a leftover from a former modfied sentence. Deleted (l. 191).

Line 186: "Somewhat" not useful, report how much δ11B was elevated in the umbilical knob

AC: The average elevation is ~0.5 ‰. Information added (l. 195).

Line 197: a total of 18 shells "were" used

AC: Corrected (l. 205).

**Review No 2 (Lubos Polerecky)**

AC: Lubos Polerecky's very helpful comments and suggestions, particularly from his analytical view, on our manuscript are very much appreciated and are all addressed in the revised manuscript (listed below).

Raitzsch et al. provide a detailed comparison between SIMS and LA-MC-ICPMS measurements of delta-11B in individual shells of benthic foraminifera. They show that intra-shell and inter-shell variability is significantly lower for the LA-based technique compared with SIMS, which they attribute to the larger volume sampled by the LA-based technique. Importantly, they show that both techniques yield very similar "average" values to those obtained by the traditional bulk measurements based on dissolved specimens. They conclude that the traditional bulk-based analysis is still the preferred approach for paleo applications, but demonstrate clearly the advantages and limits of the microanalytical techniques. The manuscript is well written and clearly organized. Also the figures are clear and of excellent quality. I only have a few minor comments and questions. I recommend the manuscript for publication after these minor issues have been resolved by the authors.

Technical comments/questions/suggestions:
l.69: Please formulate more clearly the *aim* of the study. 'What' do you want to achieve, and especially 'why'?

AC: That's right, the aim of the study was not clearly enough explained, but just vaguely outlined between l. 38-47. This is now more emphasized (l. 68-70).

l.154-155: Please clarify how this variability was calculated. Since 2*sigma is used, it may be confused with 2*sigma of the individual measurement's precision. And since permil units are used, it may be confused with the coefficient of variation (which is in percent). To avoid confusion, best would be to clarify in one sentence that 2*sigma here actually corresponds to 2*SD of n individual measurements (if I understand it correctly). Or is it 2*SE (standard error)?

AC: Yes, the reported 2*sigma variability is the 2-fold standard deviation, derived from the individual measurements. This is not to be confused with the measurement uncertainty (=precision), which is dependent on the ablation time and is also given as 2*sigma (=2*SD). So we use sigma as a statistical expression just to clarify that the SD is reported, and not the SE. We added a short notion in this sentence (l. 162).

l.157: unclear why such inistinguishability should affect variation in measured data. Please explain, or provide an alternative explanation.

AC: Right, this part is quite confusing and probably also not reflecting the truth. It is true that we observed the largest variability in the knob area that might be attributed to a signal mixture from multiple juvenile chambers, but it may also be due to the higher number of measurements compared to the other chambers. In addition, there were also similarly large variabilities found in chambers f-8 and f-9. So we have rephrased the according sentence (l. 164-166). Thanks for hinting at this inconsistency.

l.168-169: Please clarify how this was derived/deduced. Intuitively it is expected that variability in measurements is lower if larger volume is sampled. But it is unclear how you arrived to those values (e.g. ∼0.3 permil).

AC: Thanks, we missed to describe how we calculated this. We simply applied the following function to estimate the variability reduction: $u(V2) = u(V1)/sqr(V2/V1)$, where $u(V1)$ is the variability for a quoted volume, and V1 and V2 represent the two different volumes that are compared. We have added this information as a short equation in parentheses (l. 177). Irrespective of this, in section 3.1, we encountered a few slightly wrong numbers related to variability, which were corrected (see track changes), but do not affect any conclusion.

l.176/fig.4: Please clarify representation of the data in polar plots in Fig. 4. I understand that the "phi" coordinate corresponds to the chamber, but it took me a while to figure out that the r-coordinate (scale -7 to 3) corresponds to Delta-delta11B. Also I am wondering whether it would be more beneficial/transparent to show each Delta-delta-11B datapoint rather than average Delta-delta-11B deviations derived from measurements of multiple specimens. Averages may be misleading, as we know.

AC: I agree, the so-called coxcomb chart seems to be difficult to read at the beginning, but once it's understood, it is a nice way of presenting such data. In the updated figure, it should now be clearer that red represents positive and blue negative Delta-delta values. I have tried a couple graph types, also plotting all individual datapoints, as Lubos suggested, but all resulted in quite confusing graphs due to the large number of measurements (at least for SIMS). For our aim to eventually observe trends, plotting averaged deviations seems to be the catchiest way.

Did you test whether the decreasing trend between f and f-5 is significant, or you can only state "the deviation tends to decrease"?

AC: Yes, we applied the Wilcoxon-Mann-Whitney approach to test the significance of d11B differences between chambers (l. 178 ff). Also, Dennis Mayk made this a subject of discussion, and I realized that I have walked right into a trap when testing the null hypotheses. Because of the few datapoints for each chamber, I applied the statistical test on Monte-Carlo simulated d11B values (n=10) around quoted uncertainties, yielding p values smaller than 0.05, meaning that the differences in d11B between chambers f-1 and f-5 are statistically significant at a 95 % SL. This

artificially increased population size, however, led to a biased uncertainty estimation, which was also subject to papers in mathematical journals (e.g. Lin et al. (2013), Too Big to Fail: Large Samples and the p-Value Problem, http://dx.doi.org/10.1287/isre.2013.0480). If just the original data are taken into account, both the WMW and Welch t-test yield p values ~0.07, and hence the d11B differences between the chambers are not statistically different at a 95 % SL, based on the small datasets of this study. I have to apologize for this incautios and naive application of statistical tests on our data. The text is corrected in the revised manuscript (l. 186-188).

l.200: I am wondering why the authors report median instead of, for example, the mean? If it leads to a different mean in comparison to the bulk-based analysis, it should be discussed why such a difference exists. In any case, I think it needs to be clarified why median was used. Similar on l.254.

AC: The differences between medians and means are small, e.g. the SIMS median is 16.08 ‰ and the mean is 16.19 ‰, while the LA median is 15.91 ‰ and the mean is 16.17 ‰. However, the median is less sensitive against outliers than the mean and also represents the average value of a non-uniform distribution. The median is also equivalent to the average shown by the violin/box plots in Fig. 5. We have inserted a sentence why we chose the median instead of the mean (l. 206-207).

l.211: yes, I agree, but it would be useful to expand this argument towards the *source* of this variability (e.g., shell-to-shell differences in the intra-shell heterogeneity?).

AC: We think this goes slightly beyond the scope of this study, and is difficult to answer, based on our database. Maybe the range of isotopic compositions of the trigonal BO3 hosted in the calcite lattice is very large (not yet examined on the molecular scale), and hence better resolved the smaller the scale of the analytical technique is.

Figure 1: It is rather confusing to see signals for 10B and 11B centred on the same mass (10.25). Is it really so? And why? I am not familiar with the Daly detector principle.

AC: On a Nu Instruments multicollector ICP-MS, the Quad lenses are tuned in a way that the peaks of different isotopes (here 10B and 11B) coincide, i.e. the incoming ions hit the respective detectors simultaneously, where the one for high mass (11B) is on either and the low mass (10B) on the other side of the Center cup. Once the peaks coincide and the peak center is set, the information on the position of the peak center is read by the Center cup. Of course, 11B is measured on 11.009 amu and 10B on 10.013 amu, but the position of the peak center is in relation to the Center cup, and might slightly change on a daily basis, depending on the tuning parameters. We slightly modified the figure caption to clarify that the coincidence of 10B and 11B peaks appears at ~10.25 amu in the center cup.

Figure 3: Please verify the expression for 2*sigma in the graph (in red). First, the factor 1000 does not make sense if cps is in counts per second (perhaps it does if it is in kilo-counts per second). Second, if I substitute 300,000 and 300,000/4.9 for 11B and 10B, I get a factor 8.8, not 8.2. In my opinion, the formula should read as 2*sqrt(1/counts(11B)+1/counts(10B)), where counts(11B)=cps(11B)*1s*n and similarly for 10B. This is a formula for the Poisson error of 11B/10B based on counting statistics. In this formula the factor is then 0.00887 at B11=300,000. Please verify cps vs. kcps.

AC: The factor 1000 is because the boron isotopic composition is given in permil (see eq. 1 in the main text), so it's expressing the relative difference from the standard value. The measurement uncertainty (i.e. the internal precision) must hence be multiplied with 1000 to have the number in permil as well. We agree that the formula we provided is quite cumbersome, but it is mathematically correct. Based on the simpler formula provided by Lubos, we slightly modifed it to more easily enter the number of cycles (n), and multiply if with the factor 1000 to obtain the result in permil expression. The final formula is now 2*sqrt(1/cps(11B)+1/cps(10B)) / sqrt(n)*1000, which replaces the old one in the revised figure. Concerning the obtained factor at a countrate of 300,000 cps for 11B, the ratio between 11B and 10B is in nature in the order of 4, and not 4.9 (11B=80.1 %, 10B=19.9 %). Therefore, if 11B is measured at 300,000 cps, 10B is recorded at approximately 75,000 cps, thus giving a factor of 8.8 using the formula above. We added the expected counrate on 10B in the revised Fig. 3.

Editorial suggestions:
l.24: unclear why the word "presumably" is used in the abstract. It would help if the sentence is reworded to clarify what is certain and what is not (i.e., what is estimated).

AC: Right, "presumably" is a too careful term. Replaced with "estimated to be" (l. 23).

l.39: would be useful to cite few examples of such studies.

AC: These are the same references as in lines 32-33, but they are now listed here as well (l. 38-39).

l.104: perhaps it should read "45 cm3 *and* ablated"?

AC: Thanks, corrected (l. 105).

l.126: remove "and" before "that"

AC: Removed (l. 133).

In caption to Fig. 4, it should read "inset", not "inlet". Similar on l.195.

AC: Corrected, the same misspelling was in the figure captions (l. 198, 203, 219, 313, and 327).

l.184: remove "large-scale"

AC: Thanks, deleted (l. 191).

l.279: "French" - uppercase F.

AC: Corrected (l. 294).

**Interactive comment (Kaoru Kubota)**

AC: We appreciate the interactive comment of Kaoru Kubota and his careful reading of our manuscript, and will address his comments below:

Very nice work! It will be a great contribution to the community.

AC: Thank you.

Line 109: Should be 11B/10B?

AC: Well spotted! Corrected (l. 112).

Line: 184 Delete "large-scale"

AC: Thanks, deleted (l. 191).

Lines: 235-243: The readers may want to know more detail on the simulation.

Figure 7: It is interesting attempt, but I could not understand how it is simulated. If n = 4, count should be 52? (13*4) Why so much count is obtained in this simulation?

AC: Yes, it is an interesting approach, but it is not the most appropriate one, as also Dennis Mayk suggested, because it results in an underestimation of the uncertainties. We will thus apply the same R function, but on Monte Carlo simulated data sets in the revised manuscript, which gives similar results, but with correct uncertainty estimations (l. 246-258).

However, just for information on the combn() function. It uses an input population (e.g., A, B, C, D) and calculates the averages for all possible combinations among this population (k), for a given number of subsamples (n). For instance, for n=2 it calculates the averages AB, AC, AD, BC, BD, CD, so we get 6 possible combinations. To calculate the number of possible combinations (N) for any k and n, the binomial coefficient is used: $N=k!/(n!*(k-n)!)$. The possible combinations of 4 samples out of a total of 18, as in Kaoru's example, thus amount to 3060.

**II Marked-up manuscript version**

[revised manuscript text omitted]